# UNICOM: UNIVERSAL AND COMPACT REPRESENTATION LEARNING FOR IMAGE RETRIEVAL

**Xiang An**[1], **Jiankang Deng**[2] *, **Kaicheng Yang**[1], **Jiawei Li**[1], **Ziyong Feng**[1],
**Jia Guo**[3], **Jing Yang**[4], **Tongliang Liu**[5]
[1]DeepGlint, [2]Huawei, [3]InsightFace, [4]University of Cambridge, [5]University of Sydney
`xiangan@deepglint.com`,`jiankangdeng@gmail.com`

## ABSTRACT

Modern image retrieval methods typically rely on fine-tuning pre-trained encoders to extract image-level descriptors. However, the most widely used models are pre-trained on ImageNet-1K with limited classes. The pre-trained feature representation is therefore not universal enough to generalize well to the diverse open-world classes. In this paper, we first cluster the large-scale LAION 400M dataset into one million pseudo classes based on the joint textual and visual features extracted by the CLIP model. Due to the confusion of label granularity, the automatically clustered dataset inevitably contains heavy inter-class conflict. To alleviate such conflict, we randomly select partial inter-class prototypes to construct the margin-based softmax loss. To further enhance the low-dimensional feature representation, we randomly select partial feature dimensions when calculating the similarities between embeddings and class-wise prototypes. The dual random partial selections are with respect to the class dimension and the feature dimension of the prototype matrix, making the classification conflict-robust and the feature embedding compact. Our method significantly outperforms state-of-the-art unsupervised and supervised image retrieval approaches on multiple benchmarks. The code and pre-trained models are released to facilitate future research `https://github.com/deepglint/unicom`.

## 1 INTRODUCTION

Modern image retrieval methods (Lim et al., 2022; Roth et al., 2022; Kim et al., 2022; Ermolov et al., 2022; Patel et al., 2022) can be roughly decomposed into two major components: (1) the encoder (e.g., Convolutional Neural Networks (Szegedy et al., 2015; He et al., 2016) or Vision Transformer (Touvron et al., 2021; Dosovitskiy et al., 2021)) mapping the image to its compact representation and (2) the loss function (Musgrave et al., 2020) grouping the representations of similar objects while pushing away representations of dissimilar objects in the embedding space. To train the encoder, networks pre-trained on crowd-labeled datasets (e.g., ImageNet (Deng et al., 2009)) are widely used for fine-tuning (Wang et al., 2019; Kim et al., 2021). However, ImageNet only contains 1,000 pre-defined object classes. The feature representation learned from ImageNet is not universal enough to generalize to diverse open-world objects.

Even though fully supervised pre-training can benefit from a strong semantic learning signal for each training example, supervised learning is not scalable because manual annotation of large-scale training data is time-consuming, costly, and even infeasible. By contrast, self-supervised pre-training methods (He et al., 2020; 2022; Radford et al., 2021; Jia et al., 2021) can be easily scaled to billions of unlabeled examples by designing an appropriate pretext task, such as solving jigsaw puzzles (Noroozi & Favaro, 2016), invariant mapping (Chen & He, 2021), and image-text matching (Radford et al., 2021; Jia et al., 2021). Among them, CLIP (Radford et al., 2021) has recently demonstrated success across various downstream tasks (e.g., image retrieval and classification) due to superior feature representation empowered by image-text contrastive learning. Specifically, CLIP aligns the visual and textual signals of each instance into a unified semantic space by cross-modal instance

---
*denotes corresponding author.

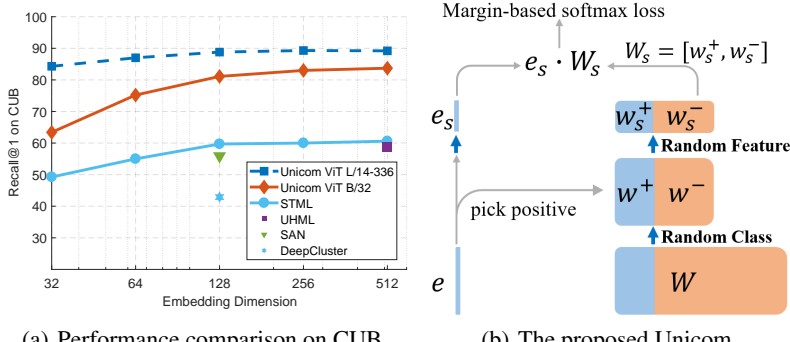

(a) Performance comparison on CUB      (b) The proposed Unicom

Figure 1: (a) Accuracy in Recall@1 versus embedding dimension on the CUB dataset. The proposed Unicom is only trained on the LAION 400M dataset without any manual annotation. (b) The proposed Unicom employs two random selections along the class dimension and the feature dimension to alleviate inter-class conflict and achieve compact representation, respectively.

discrimination. Nevertheless, the instance discrimination method used by CLIP can hardly encode the semantic structure of training data, because instance-wise contrastive learning always treats two samples as a negative pair if they are from different instances, regardless of their semantic similarity. When thousands of instances are selected into the training batch to form the contrastive loss, negative pairs that share similar semantics will be undesirably pushed apart in the embedding space.

To handle the limitations of instance discrimination, cluster discrimination has been proposed for deep unsupervised learning through jointly learning image embeddings and cluster assignments. Learning representations with clusters will pull similar instances together, which is beneficial for capturing semantic structures in data. DeepCluster (Caron et al., 2018) performs iterative clustering by k-means and classification by cross-entropy loss, while SeLa (Asano et al., 2020) proposes to solve an optimal transport problem for balanced assignment. However, both DeepCluster and SeLa need labels to be assigned offline in a batch mode with representations of all instances. To reduce the cost of batch mode clustering, ODC (Zhan et al., 2020), SwAV (Caron et al., 2020), and CoKe (Qian et al., 2022) apply online clustering to avoid the multiple iterations over the entire dataset. Despite improved efficiency, the online clustering method still suffers from the collapsing problem (i.e., a dominant cluster includes instances from multiple classes or most of the instances).

In this paper, we aim at boosting the semantic embedding power of the CLIP model by introducing a novel cluster discrimination approach. We first conduct one step of off-line clustering by using the image and text features from a pre-trained CLIP model (Radford et al., 2021). Due to the limited discrimination power of the CLIP model, the pseudo clusters contain heavy inter-class conflict. Instead of optimizing the cluster assignment (Qian et al., 2022), we focus on how to robustly train a classifier on the automatically clustered large-scale data. More specifically, we explore two random selections on the prototype matrix $W \in R^{d \times k}$ when preparing the classification loss (as illustrated in Fig. 1(b)). The first one is with respect to the class dimension ($k$), and only part of negative prototypes are selected for inter-class comparisons, which helps alleviate inter-class conflict. The second one is with respect to the feature dimension ($d$), and only part of features are randomly selected to construct the classification loss, enhancing the representation power of each neuron and making feature representation compact for the efficient image retrieval task. Concurrent with our work, partial selection mechanisms along class and feature are separately proposed in (An et al., 2021; 2022) and (Xu et al., 2022) to accelerate model training and enhance locally distinguishable features. Both of their experiments are conducted on cleaned face recognition datasets. By contrast, we target at universal and compact representation learning from automatically clustered large-scale data. The main contributions of our paper are the following:

- We propose a novel cluster discrimination method for universal and compact representation learning. In the clustering step, we employ both image and text features from the pre-trained CLIP model. In the discrimination step, we explore two random selections along class and feature, which can potentially alleviate inter-class conflict and improve the feature compactness, respectively.

- For both zero-shot learning tasks (e.g., linear probe and unsupervised image retrieval) and transfer learning tasks (e.g., ImageNet-1K classification and supervised image retrieval), the proposed random negative prototype selection for conflict-robust cluster discrimination can significantly boost the representation power compared to the instance discrimination based model (e.g., CLIP).

## 2 RELATED WORK

**Visual Model Pre-training.** Model pre-training for visual recognition can be categorized into three main groups: (1) supervised pre-training on datasets with manually annotated class labels (e.g., ImageNet-1K/-21K (Deng et al., 2009) and JFT-300M/-3B (Dosovitskiy et al., 2021; Zhai et al., 2022)), (2) weakly-supervised pre-training by using hashtag (Mahajan et al., 2018; Singh et al., 2022) or text descriptions (Radford et al., 2021; Jia et al., 2021), and (3) unsupervised pre-training (Chen et al., 2020; He et al., 2020; Caron et al., 2018). Since supervised pre-training relies on expensive manual annotations, we focus on annotation-free pre-training which has the advantages of being easily scaled to billions of training images and being able to learn universal feature representations for downstream tasks.

**Instance and Cluster Discrimination.** Instance discrimination (Chen et al., 2020; He et al., 2020; Radford et al., 2021) is realized with a contrastive loss which targets at pulling closer samples from the same instance while pushing away samples from different instances. Despite the impressive performance, instance-wise contrastive learning can not capture the semantic information from the training data because it is trained to ignore the similarity between different instances. Cluster discrimination (Caron et al., 2018; Zhan et al., 2020; Li et al., 2020a) is processed with iterative steps: the clustering step to assign pseudo class labels for each sample, and then the classification step to map each sample to its assigned label. Since one cluster has more than one instance, learning representations with clusters will gather similar instances together, which can explore potential semantic structures in data. As a representative work, DeepCluster (Caron et al., 2018) adopts a standard k-means for clustering, but it contains degenerate solutions. To this end, recent research work (Asano et al., 2020; Caron et al., 2020; Qian et al., 2022) focuses on improving the label assignment during clustering but employs a standard cross-entropy loss during discrimination. In this paper, we only employ one step of off-line clustering but design a robust classifier to achieve good feature representation when training on the automatically clustered large-scale data.

**Image Retrieval.** Image retrieval task typically relies on fine-tuning pre-trained visual models (Szegedy et al., 2015; He et al., 2016; Dosovitskiy et al., 2021) and can be divided into two learning categories: supervised and unsupervised metric learning. For supervised metric learning, pair-wise loss (Hadsell et al., 2006; Schroff et al., 2015; Sohn, 2016) and cross-entropy loss (Zhai & Wu, 2019; Deng et al., 2019; Sun et al., 2020; Qian et al., 2019) are extensively studied and recent bench-marking results (Musgrave et al., 2020) indicate that the margin-based softmax loss (e.g., ArcFace (Deng et al., 2019)) can achieve state-of-the-art performance. For unsupervised metric learning, pseudo labeling methods are employed to discover pseudo classes by applying k-means clustering (Kan et al., 2021; Li et al., 2020b), hierarchical clustering (Yan et al., 2021), random walk (Iscen et al., 2018), and class-equivalence relations (Kim et al., 2022) to unlabeled training data. In this paper, we focus on universal and compact feature embedding for both unsupervised and supervised image retrieval task.

## 3 METHODOLOGY

### 3.1 PRELIMINARIES OF INSTANCE AND CLUSTER DISCRIMINATION

Given a training set $X = \{x_1, x_2, ..., x_n\}$ including $n$ images, feature representation learning aims at learning a function $f$ that maps images $X$ to embeddings $E = \{e_1, e_2, ..., e_n\}$ with $e_i = f(x_i)$, such that embeddings can describe the similarities between different images.

Instance discrimination achieves this objective by optimizing a contrastive loss function defined as:

$$\mathcal{L}_{\text{instance}} = -\sum_{i=1}^{n} \log \frac{\exp(e_i'^T e_i)}{\sum_{j=0}^{m} \exp(e_j'^T e_i)}, \tag{1}$$

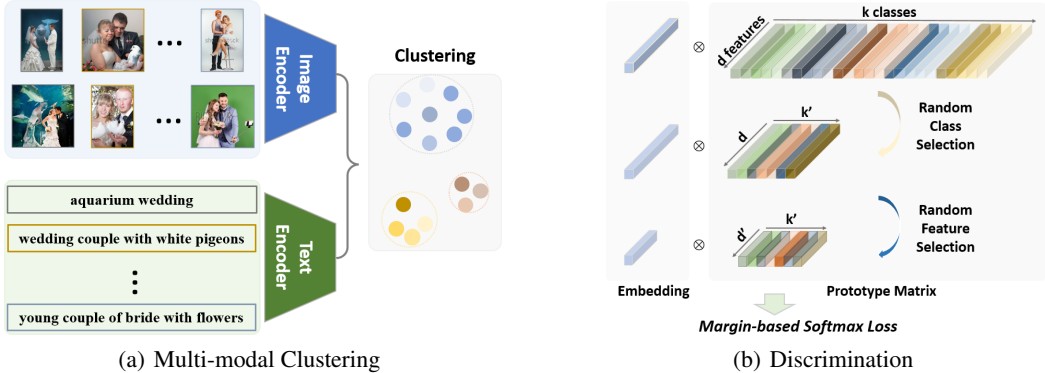

(a) Multi-modal Clustering                    (b) Discrimination

Figure 2: Illustration of the proposed method. (a) The multi-modal clustering includes one off-line step of k-means on features from image and text produced by a pre-trained CLIP model (Radford et al., 2021). (b) Using the assigned clusters as pseudo-labels, we propose a conflict-robust and representation-compact classification method through random class and feature selection along the two dimensions of the prototype matrix.

where $e_i$ and $e'_i$ are positive embeddings of the instance $i$, and $e'_j$ consists of one positive embedding of $i$ and its $m$ negative embeddings from other instances.

By contrast, cluster discrimination for representation learning consists of two main phases: clustering and discrimination. The clustering step assigns each instance a pseudo class label that will be subsequently used as supervision to train a classifier in the discrimination step. Following this, automatic clustering on the features $e_i = f(x_i)$ is first performed to obtain $k$ clusters and the centroid $w_i$ is viewed as the prototype of $i$-th cluster. Then, the training data $\{x_i\}_{i=1}^n$ are partitioned into $k$ classes represented by prototypes $W = \{w_i\}_{i=1}^k$. With pseudo labels and centroids obtained from the clustering step, cluster discrimination can be implemented by optimizing a standard softmax classification loss as:

$$\mathcal{L}_{\text{cluster}} = -\sum_{i=1}^n \log \frac{\exp(w_i^T e_i)}{\sum_{j=1}^k \exp(w_j^T e_i)}, \qquad (2)$$

where $e_i$ is the embedding of the image $x_i$ and $x_i$ belongs to the class represented by $w_i$. By comparing Eq. 1 and Eq. 2, we can observe the difference that instance discrimination employs an augmented feature $e'_i$ to calculate the similarities while cluster discrimination uses a prototype $w_i$.

## 3.2 MULTIMODAL CLUSTERING

In this paper, we focus on the standard clustering algorithm, $k$-means, which takes a set of vectors as input and clusters them into $k$ distinct groups based on the nearest neighbor criterion. To seek a better representation, we combined the image and text features produced by the pre-trained CLIP model (Radford et al., 2021) due to their mutual complementary nature. The clustering step jointly learns a $d \times k$ centroid matrix $W$ and the cluster assignments $y_i$ of each image $x_i$ by solving the following problem:

$$\min_{W \in \mathbb{R}^{d \times k}} \frac{1}{n} \sum_{i=1}^n \min_{y_i \in \{0,1\}^k} \|\Phi(f(x_i), f'(x'_i)) - W y_i\|_2^2 \quad \text{s.t.} \quad y_i^\top \mathbf{1_k} = \mathbf{1}, \qquad (3)$$

where $f(x_i)$ is the image feature embedding by the image encoder $f$ and $f'(x'_i)$ is the text feature embedding by the text encoder $f'$, $\Phi$ is a feature ensemble function, $W \in R^{d \times k}$ is the centroid matrix, $y_i$ in $\{0,1\}^k$ is a single label assignment constrained by $y_i^\top \mathbf{1_k} = \mathbf{1}$, and $\mathbf{1_k}$ is 1-vector of size $k$. In this work, we employ the simplest feature ensemble function, that is, averaging the image and text features, as the CLIP model provides an aligned visual-textual representation.

Considering that iterative clustering and discrimination are time-consuming, we only employ one step of off-line clustering in this work. Aided by the efficient feature quantization (Johnson et al., 2019), the large-scale LAION 400M dataset can be clustered within 10 minutes using the embedded image and text features. The only hyper-parameter we consider here is the cluster number $k$. Even though the clustering step is straightforward, the automatically clustered large-scale dataset inevitably

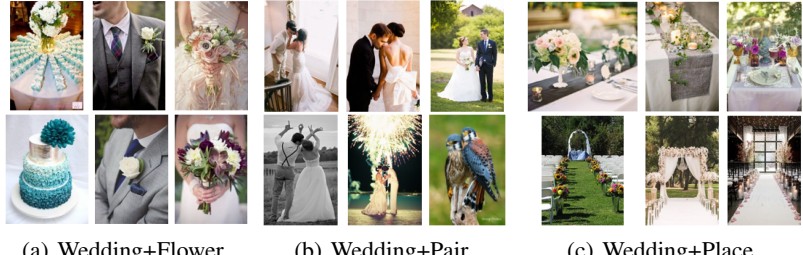

| (a) Wedding+Flower | (b) Wedding+Pair | (c) Wedding+Place |

Figure 3: Inter-class conflict between the automatically-clustered classes. The class name is given based on the observation of images and texts. Inter-class conflict exists due to specific granularity definitions and multi-label signals in one image.

confronts inter-class conflict due to multi-label signals in one image and specific definition of class granularity (as illustrated in Fig. 3). For instance, the bird pair in Fig. 3(b) is clustered into wedding pair, which will be conflicted with another specific category of bird. In addition, the close-up capture of wedding flowers in Fig. 3(a) also exists in the class of wedding place, where flowers are the most popular decoration.

### 3.3 CONFLICT-ROBUST AND REPRESENTATION-COMPACT DISCRIMINATION

After the clustering step, we can employ a vanilla classification loss (e.g., softmax) to learn the feature representation. For the softmax loss given in Eq. 2, the derivatives to a class-wise prototype $w_j \in \mathbb{R}^d$ and a sample embedding feature $e_i \in \mathbb{R}^d$ are:

$$\frac{\partial \mathcal{L}}{\partial w_j} = \sum_{i=1}^{b}(p_{ij} - \mathbb{1}\{y_i == j\})e_i, \qquad \frac{\partial \mathcal{L}}{\partial e_i} = \sum_{j=1}^{k}(p_{ij} - \mathbb{1}\{y_i == j\})w_j, \qquad (4)$$

where $b$ is the batch size, $k$ is the class number, and $p_{ij} = e^{w_j^T e_i}/\sum_{l=1}^{k} e^{w_l^T e_i}$ is the normalized probability of the sample embedding $e_i$ belonging to the prototype $w_j$, $\mathbb{1}(\cdot)$ is the indicator function which is $1$ when the statement is true and $0$ otherwise. In Eq. 4, the derivative of the prototype is a "weighted sum" over sample features from the mini-batch, and the derivative of the sample feature is a "weighted sum" over the prototypes of all classes. If conflicted classes exist, wrong gradient signals from these conflicted negative prototypes will affect the update of model parameters.

To this end, we propose a random negative prototype selection to efficiently construct a negative prototype subset from the entire negative prototypes. Therefore, the derivative to a sample embedding feature is:

$$\frac{\partial \mathcal{L}}{\partial e_i} = -((1 - p^+)w^+ - \sum_{j \in \mathbb{S}, j \neq y_i} p_j^- w_j^-), \qquad (5)$$

where $p^+$ and $w^+$ denote the probability and prototype of the positive class, $p_j^-$ and $w_j^-$ refer to negative probabilities and prototypes, $\mathbb{S}$ is a subset of all negative classes and one positive class, $|\mathbb{S}| = k * r_1$, and $r_1 \in [0, 1]$ is the sampling ratio. Even though all class-wise prototypes are still maintained throughout the whole training process, only positive prototypes and a subset of negative prototypes are selected and updated in each iteration. Therefore, the inter-class conflict will be reduced as the possibility of sampling a conflict negative prototype is directly decreased by $r_1$.

To achieve a compact representation for efficient image retrieval, previous methods (Babenko & Lempitsky, 2015; Tolias et al., 2016) adopt Principal Component Analysis (PCA) on an independent set for dimension reduction. To reduce the descriptor dimension to $d'$, only eigenvectors corresponding to $d'$ largest eigenvalues are retained. However, the representation produced by PCA is sub-optimal because it is a post-processing step detached from the target task. To this end, we propose a feature approximation strategy by randomly selecting subspace features to construct the classification loss:

$$\mathcal{L}_{\text{unicom}} = -\sum_{i=1}^{n} \log \frac{\exp((\Gamma_t \odot w_i)^T(\Gamma_t \odot e_i))}{\sum_{j \in \mathbb{S}} \exp((\Gamma_t \odot w_j)^T(\Gamma_t \odot e_i))}, \qquad (6)$$

where $\Gamma_t \in \{0, 1\}^d$ is a random binary vector at the iteration step of $t$, the non-zero element ratio of $\Gamma_t$ is $r_2 \in [0, 1]$, $\odot$ denotes element-wise product. Different from the well-known regularization

technique, Dropout (Srivastava et al., 2014), our random feature selection $\Gamma_t$ is same for all training samples within the mini-batch. $\Gamma_t$ is applied to both feature $e_i$ and prototypes $w_j$, thus the dimension of derivatives in Eq. 4 decreases to $d'$. By contrast, Dropout is independently applied to each individual feature $e_i$ within the mini-batch by setting a specific ratio $r_3 \in [0, 1]$ of neurons to 0 and enlarging the rest neurons by $1/(1 - r_3)$. The dimension of derivatives in Eq. 4 is still $d$. Therefore, the sub-feature optimization in the proposed random feature selection can not be completed by directly calling the Dropout function. Since the binary vector varies at different iterations, different sub-features are trained and sub-gradients are calculated. This leads to a solution that each embedding neuron contains the similarity representation power.

The schematic of the proposed method is in Fig. 2(b). As shown, the prototype matrix $W$ is maintained in the memory at the dimension of $d \times k$ during the whole training process, but only part of the classes ($k' = k * r_1$) and features ($d' = d * r_2$) are randomly selected to construct the softmax loss. The first random selection along the class dimension is for conflict-robust learning to achieve universal representation and the second random selection along the feature dimension is for feature compression required by efficient retrieval. Therefore, we name our method UNIversal and COMpact (UNICOM) representation learning.

## 4 EXPERIMENTS

### 4.1 IMPLEMENTATION DETAILS

Unless otherwise specified, all ViT models in our experiments follow the same architecture designs in CLIP, and are trained from scratch for 32 epochs on the automatically clustered LAION 400M dataset (Section 3.2) with cluster number $k = 1M$. During training, we randomly crop and horizontally flip each image to get the input image with $224 \times 224$ resolution. We set the random class sampling ratio $r_1$ as 0.1 in the pre-training step. The training is conducted on 128 NVIDIA V100 GPUs across 16 nodes. To save memory and scale up the batch size, mixed-precision and gradient checkpoint are used. We use AdamW (Loshchilov & Hutter, 2018) as the optimizer with an initial learning rate of 0.001, and a weight decay of 0.05. We employ margin-based softmax loss, ArcFace (Deng et al., 2019; 2020), for both pre-training and image retrieval tasks. The margin value is set to 0.3 and the feature scale is set to 64. For supervised retrieval, we follow the data-split settings of the baseline methods (Patel et al., 2022; Ermolov et al., 2022) to fine-tune models.

### 4.2 COMPARISONS ON FEATURE REPRESENTATION LEARNING

In this section, we first compare the performance of the proposed method and other baseline models (i.e., CLIP and OPEN-CLIP) on the linear probe and unsupervised image retrieval. Specifically, after the training on the automatically clustered 1M classes, we fix the backbones of our models. For the linear probe task, we learn an additional FC layer for classification on each test set. For unsupervised image retrieval, we directly use the embedding features for testing. Then, we fine-tune the pre-trained models for supervised image retrieval on each image retrieval dataset.

**Linear Probe.** Following the same evaluation setting as CLIP (Radford et al., 2021), we freeze the pre-trained models on LAION 400M dataset and only fine-tune the last linear classification layer. We report the linear probing performance over 13 datasets in Tab. 1. The proposed conflict-robust cluster discrimination method significantly outperforms the CLIP and OPEN-CLIP (Ilharco et al., 2021) models. Notably, our ViT B/32, ViT B/16, and ViT L/14 models surpass counterparts of OPEN-CLIP by 3.6%, 2.7% and 1.4% on average with the same training data, indicating that the proposed cluster discrimination can enhance the representation power over instance discrimination.

**Unsupervised Image Retrieval.** In Tab. 2, we compare the performance of unsupervised image retrieval by directly using the pre-trained models for feature embedding. The GLDv2 (Weyand et al., 2020) employs mean Average Precision@100 (mAP@100) as the evaluation metric, while other datasets use Recall@1. Our ViT L/14 model achieves 69.9% average result across 7 image retrieval datasets, surpassing the OPEN-CLIP counterpart by 7.5% and even outperforming the larger OPEN-CLIP model ViT H/14 by 5.4%. The reason behind this significant improvement is that the proposed cluster discrimination can capture the semantic structure in data, which is crucial for the image retrieval task. In Fig. 1(a), we compare our method with the state-of-the-art unsupervised image retrieval approach, STML (Kim et al., 2022), under different dimension constraints on the

Table 1: Top-1 accuracy(%) of linear probe on 13 image classification datasets. The proposed cluster discrimination significantly outperforms OPEN-CLIP (Ilharco et al., 2021) on average by using the same training data (i.e., LAION 400M). "CLIP-R" denotes testing the public CLIP-ViT models in our code base. "-336" refers to one additional epoch of pre-training at a higher $336 \times 336$ resolution to boost performance.

| | | CIFAR10 | CIFAR100 | Caltech101 | Cars | Flowers | Food101 | Birdsnap | SUN397 | DTD | Aircraft | Pets | EuroSAT | ImageNet | Average |
|---|---|---|---|---|---|---|---|---|---|---|---|---|---|---|---|
| CLIP | ViT B/32 | 95.1 | 80.5 | 93.0 | 81.8 | 96.9 | 88.8 | 58.5 | 76.6 | 76.5 | 52.0 | 90.0 | 97.0 | 76.1 | 81.8 |
| | ViT B/16 | 96.2 | 83.1 | 94.7 | 86.7 | 98.1 | 92.8 | 67.8 | 78.4 | 79.2 | 59.5 | 93.1 | 97.1 | 80.2 | 85.1 |
| | ViT L/14 | 98.0 | 87.5 | 96.5 | 90.9 | 99.2 | 95.2 | 77.0 | 81.8 | 82.1 | 69.4 | 95.1 | 98.2 | 83.9 | 88.8 |
| | ViT L/14-336 | 97.9 | 87.4 | 96.0 | 91.5 | 99.2 | 95.9 | 79.9 | 82.2 | 83.0 | 71.6 | 95.1 | 98.1 | 85.4 | 89.5 |
| CLIP-R | ViT B/32 | 96.0 | 82.5 | 94.1 | 86.0 | 97.8 | 92.7 | 61.1 | 79.1 | 78.4 | 58.9 | 93.0 | 95.3 | 75.3 | 83.9 |
| | ViT B/16 | 96.0 | 82.5 | 94.1 | 86.0 | 97.8 | 92.7 | 69.5 | 79.1 | 78.4 | 58.9 | 93.0 | 95.3 | 79.6 | 84.8 |
| | ViT L/14 | 98.1 | 87.2 | 96.0 | 90.7 | 99.2 | 95.3 | 77.8 | 81.5 | 80.9 | 68.0 | 94.9 | 96.7 | 84.1 | 88.5 |
| | ViT L/14-336 | 97.8 | 87.1 | 96.3 | 91.4 | 99.2 | 95.9 | 80.9 | 82.2 | 82.4 | 71.2 | 95.1 | 96.8 | 84.9 | 89.3 |
| OPEN | ViT B/32 | 95.3 | 82.2 | 93.3 | 87.5 | 96.5 | 86.2 | 61.4 | 75.3 | 78.8 | 52.4 | 88.0 | 96.5 | 73.8 | 82.1 |
| | ViT B/16 | 96.4 | 84.0 | 94.1 | 91.8 | 98.1 | 90.7 | 71.2 | 78.7 | 81.6 | 59.3 | 90.0 | 96.2 | 78.5 | 85.4 |
| | ViT L/14 | 97.9 | 87.9 | 95.5 | 93.6 | 98.8 | 93.3 | 78.0 | 81.0 | 83.0 | 64.4 | 93.3 | 97.1 | 81.5 | 88.1 |
| Ours | ViT B/32 | 96.8 | 86.6 | 94.6 | 93.3 | 98.5 | 85.8 | 70.2 | 74.6 | 78.0 | 70.7 | 93.1 | 96.8 | 75.0 | 85.7 |
| | ViT B/16 | 97.3 | 87.7 | 95.1 | 94.3 | 98.9 | 91.2 | 79.3 | 77.1 | 81.2 | 73.4 | 93.9 | 97.0 | 79.1 | 88.1 |
| | ViT L/14 | 98.5 | 90.8 | 95.7 | 94.6 | 99.3 | 93.4 | 82.4 | 80.0 | 82.2 | 74.5 | 94.2 | 96.7 | 81.8 | 89.5 |
| | ViT L/14-336 | 98.5 | 90.7 | 95.7 | 95.1 | 99.4 | 94.3 | 85.1 | 79.7 | 82.0 | 78.1 | 94.5 | 97.2 | 82.7 | 90.2 |

Table 2: Performance of unsupervised image retrieval on 7 image retrieval datasets. The proposed conflict-robust cluster discrimination significantly outperforms OPEN-CLIP on average by using the same training data.

| | | CUB | Cars | SOP | In-Shop | INaturalist | VehicleID | | | GLDv2 | | Average |
|---|---|---|---|---|---|---|---|---|---|---|---|---|
| | | | | | | | Small | Medium | Large | Private | Public | |
| CLIP | B/32 | 56.7 | 79.0 | 60.5 | 45.4 | 53.0 | 54.8 | 52.2 | 44.6 | 7.5 | 7.5 | 46.1 |
| | B/16 | 66.1 | 85.2 | 63.2 | 56.1 | 63.1 | 55.1 | 50.9 | 43.8 | 8.4 | 10.6 | 50.3 |
| | L/14 | 76.0 | 90.3 | 65.6 | 62.9 | 72.9 | 62.4 | 58.9 | 51.8 | 12.1 | 13.6 | 56.7 |
| | L/14-336 | 77.3 | 90.9 | 67.8 | 66.3 | 76.8 | 64.1 | 60.3 | 53.8 | 17.0 | 15.6 | 59.0 |
| OPEN | B/32 | 62.3 | 89.2 | 65.9 | 64.6 | 54.9 | 71.0 | 67.1 | 59.9 | 9.17 | 8.4 | 55.2 |
| | B/16 | 71.4 | 92.9 | 68.7 | 74.2 | 64.1 | 73.3 | 70.1 | 63.7 | 12.1 | 11.0 | 60.2 |
| | L/14 | 79.4 | 94.9 | 70.6 | 77.1 | 71.0 | 72.0 | 69.1 | 62.0 | 14.5 | 13.8 | 62.4 |
| | H/14 | 83.1 | 95.7 | 72.7 | 78.8 | 77.0 | 72.7 | 69.7 | 61.9 | 17.7 | 15.3 | 64.5 |
| Ours | B/32 | 83.7 | 95.9 | 70.0 | 72.8 | 64.6 | 74.9 | 72.0 | 65.4 | 15.1 | 13.3 | 62.8 |
| | B/16 | 86.5 | 96.8 | 70.4 | 74.6 | 73.6 | 74.5 | 70.6 | 58.7 | 18.7 | 17.2 | 64.2 |
| | L/14 | 88.5 | 96.9 | 72.7 | 83.6 | 77.1 | 83.7 | 80.2 | 74.6 | 21.1 | 20.1 | 69.9 |
| | L/14-336 | 89.2 | 97.3 | 74.5 | 86.7 | 81.0 | 84.1 | 81.4 | 75.6 | 23.2 | 22.0 | 71.5 |

CUB dataset. We set the random feature selection ratio $r_2$ as 0.5 for one additional training epoch on the LAION 400M dataset. Then, we select the first 256-D, 128-D, 64-D, and 32-D features for testing. STML employs an ImageNet-1K pre-trained GoogleNet (Szegedy et al., 2015) and then explores unsupervised training on the CUB dataset. Even though our ViT-based model is only trained on the automatically clustered LAION 400M dataset without any further training on the image retrieval dataset, our method outperforms STML (Kim et al., 2022) by a large margin across different test dimensions, indicating the superiority of the proposed random feature selection for compact feature representation learning.

**Fine-tune for ImageNet-1K Classification.** In Tab. 3, we compare our method with state-of-the-art supervised and weakly supervised pre-training (Dosovitskiy et al., 2021; Zhai et al., 2022; Singh et al., 2022) in transfer-learning experiments on ImageNet-1k. Our models consistently outperform OPEN-CLIP models in the Top-1 accuracy, showing the superiority of the proposed cluster discrimination. For ViT B/16, our pre-training achieves 85.9%, surpassing both the supervised pre-training on IN-21K (84.0%) and the weakly supervised pre-training on IG 3.6B (85.3%). In addition, our ViT L/14 obtains 88.3%, outperforming ViT L/16 pre-trained on JFT 300M (87.8%) and ViT L/16 pre-trained on IG 3.6B (88.1%). The overall results in ImageNet-1K classification task show that our models are

Table 3: Transfer-learning accuracy of models pre-trained on the specified dataset followed by fine-tuning and testing on ImageNet.

| Model | Pre-training Dataset | Resolution | | IN-1K Top-1 Accuracy | FLOPs (B) |
|---|---|---|---|---|---|
| | | Pretrain | Finetune | | |
| *Supervised pre-training* | | | | | |
| ViT L/32 (Dosovitskiy et al., 2021) | IN-21k | 224 | 384 | 81.3 | 54.4 |
| ViT B/16 (Dosovitskiy et al., 2021) | IN-21k | 224 | 384 | 84.0 | 55.6 |
| ViT L/16 (Dosovitskiy et al., 2021) | IN-21k | 224 | 384 | 85.2 | 191.5 |
| ViT L/16 (Dosovitskiy et al., 2021) | JFT 300M | 224 | 512 | 87.8 | 362.9 |
| ViT L/16 (Zhai et al., 2022) | JFT 3B | 224 | 384 | 88.5 | 191.5 |
| *Weakly supervised pre-training* | | | | | |
| ViT B/16 (Singh et al., 2022) | IG 3.6B | 224 | 384 | 85.3 | 55.6 |
| ViT L/16 (Singh et al., 2022) | IG 3.6B | 224 | 512 | 88.1 | 362.9 |
| ViT B/32 OPEN-CLIP | LAION 400M | 224 | 384 | 83.0 | 15.5 |
| ViT B/16 OPEN-CLIP | LAION 400M | 224 | 384 | 85.4 | 55.6 |
| ViT L/14 OPEN-CLIP | LAION 400M | 224 | 518 | 87.7 | 507.8 |
| ViT B/32 Ours | LAION 400M | 224 | 384 | 83.6 | 15.5 |
| ViT B/16 Ours | LAION 400M | 224 | 384 | 85.9 | 55.6 |
| ViT L/14 Ours | LAION 400M | 224 | 518 | 88.3 | 507.8 |

Table 4: Performance of supervised image retrieval on 7 image retrieval datasets.

| | ViT-B/32 | ViT-B/16 | ViT-L/14 | ViT-L/14-336 | Previous SOTA |
|---|---|---|---|---|---|
| CUB | 85.8 | 88.8 | 89.7 | 90.1 | 85.6 ViT-S/16 (Ermolov et al., 2022) |
| Cars | 97.3 | 97.7 | 97.9 | 98.2 | 94.8 SE-ResNet-50 (Jun et al., 2019) |
| SOP | 87.1 | 88.8 | 89.9 | 91.2 | 88.0 ViT-B/16 (Patel et al., 2022) |
| In-Shop | 94.8 | 95.5 | 96.0 | 96.7 | 92.7 ViT-S/16 (Ermolov et al., 2022) |
| INaturalist | 72.8 | 82.5 | 85.4 | 88.9 | 83.9 ViT-B/16 (Patel et al., 2022) |
| VehicleID-Small | 95.4 | 96.4 | 96.5 | 97.0 | 96.2 ViT-B/16 (Patel et al., 2022) |
| VehicleID-Medium | 94.1 | 95.1 | 95.7 | 96.1 | 95.2 ViT-B/16 (Patel et al., 2022) |
| VehicleID-Large | 93.6 | 95.0 | 95.4 | 96.0 | 94.7 ViT-B/16 (Patel et al., 2022) |
| GLDv2-Private | 32.6 | 35.7 | 36.1 | 36.4 | 32.5 ResNet101 (Lee et al., 2022) |
| GLDv2-Public | 29.7 | 32.4 | 33.0 | 33.1 | 24.6 ResNet50 (Tan et al., 2021) |

very competitive as they can achieve better or comparable accuracy even though the training data used by the competitors are much larger (e.g., JFT 3B and IG 3.6B).

**Fine-tune for Supervised Image Retrieval.** In Tab. 4, we compare the proposed approach with the latest image retrieval methods (Patel et al., 2022; Ermolov et al., 2022) trained with vision transformer. During fine-tuning of our models, the random negative class selection ratio $r_1$ is set to 1.0 as the training data is clean. Under different computing regimes, the proposed method consistently surpasses RA@K (Patel et al., 2022) on the SOP, iNaturalist, and VehicleID datasets and outperforms Hyp-ViT (Ermolov et al., 2022) on the CUB and In-shop datasets.

## 4.3 ABLATION STUDY

**Encoder for Clustering.** In Tab. 5, we compare the results of linear probe and unsupervised image retrieval under image-based clustering and text-based clustering by using the pre-trained CLIP and OPEN-CLIP models. As we can see, the text encoder is more powerful than the image encoder, and image and text signals are complementary as the joint clustering significantly outperforms each individual. By referring to Tab. 1 and Tab. 2, the OPEN-CLIP ViT B/32 model achieves $82.1\%$ and $55.2\%$ average results on the linear probe and unsupervised image retrieval tasks, while the proposed cluster discrimination method ($r_1 = 0.1$) significantly boosts the performance to $84.1\%$ and $61.1\%$ by using the OPEN-CLIP image and text models for clustering. By using the CLIP image and text models for clustering, the performance can further increase to $85.7\%$ and $62.8\%$ on the linear probe and unsupervised image retrieval tasks. Therefore, we choose the CLIP models for clustering.

**Cluster Number.** In Tab. 5, we also compare the performance under different cluster numbers, i.e., 100K, 1M, and 10M, by using the CLIP image and text models. As can be seen, the best results can be achieved when the cluster number is set as 1 million, with the average image number per class being

Table 5: Ablation study on multi-modal clustering. ViT B/32 is used here for model training on the LAION 400M dataset, which is automatically clustered by different pre-trained models. We report the average performance of linear probe and unsupervised image retrieval.

| Tasks | CLIP | | | OPEN-CLIP | | | Cluster Number by CLIP | | |
|---|---|---|---|---|---|---|---|---|---|
| | Image | Text | Joint | Image | Text | Joint | 100K | 1M | 10M |
| Linear Probe | 84.4 | 85.3 | 85.7 | 83.9 | 84.0 | 84.1 | 75.9 | 85.7 | 83.6 |
| Unsup. Retr. | 61.8 | 62.3 | 62.8 | 58.9 | 60.1 | 61.1 | 53.2 | 62.8 | 60.7 |

Table 6: Ablation study on random negative class selection and random feature selection. ViT-B/32 is used here and we report the average performance of linear probe and unsupervised image retrieval.

| Tasks | Random Class Ratio ($r_1$) | | | | Random Feature Ratio ($r_2$) | | | Dropout Ratio ($r_3$) | |
|---|---|---|---|---|---|---|---|---|---|
| | 0.05 | 0.1 | 0.3 | 1.0 | 1.0 | 0.5 | 0.25 | 0.25 | 0.5 |
| Linear Probe | 85.1 | 85.7 | 84.9 | 77.9 | 85.7 | 85.5 | 84.2 | 85.4 | 85.1 |
| Unsup. Retr. | 62.3 | 62.8 | 62.1 | 55.9 | 62.8 | 62.7 | 62.0 | 62.5 | 62.3 |
| Unsup. Retr. [256] | - | - | - | - | 61.4 | 61.8 | 61.0 | 60.7 | 60.1 |

around 400. The cluster number needs to be balanced between the intra-class noises and inter-class noises. Too small cluster numbers will incur heavy intra-class noise, which dramatically decreases the performance of the pre-trained classification model. Besides, too many clusters will increase the computation and storage burden on the FC layer. Most important, the over-decomposition will increase the inter-class noise ratio and undermine the discriminative power of the pre-trained model.

**Random Class Selection.** In Tab. 6, we train ViT B/32 models under different inter-class sampling ratios. The basic margin-based softmax loss ($r_1 = 1.0$) only achieves 77.9% on the linear probe task as it can hardly adapt to the heavy inter-class conflict in the automatically clustered dataset. When the sampling ratio is decreased from 1.0 to 0.3 and 0.1, our method exhibits consistently better performance than the baseline, indicating random inter-class sampling is beneficial for the model's robustness. When $r_1$ is set to 0.05, there is a slight performance drop because the inter-class interaction is insufficient during training. Therefore, we choose the random negative class selection ratio as 0.1, obtaining 85.7% and 62.8% on the linear probe and unsupervised image retrieval tasks.

**Random Feature Selection.** In Tab. 6, we compare the performance of the proposed Unicom under different random feature selection ratios ($r_2$) on the task of dimension-constrained unsupervised image retrieval. Here, we also include Dropout at different drop ratios ($r_3$) for comparison. From the results, we can have the following observations: (1) both random feature selection and Dropout can not improve linear probe and unsupervised image retrieval at a full dimension of 512 as the LAION 400M dataset is large enough and regularization is not necessary for the final classification layer, (2) there is slight performance drop when the random feature selection ratio is decreasing, and (3) the proposed random feature selection ($r_2 = 0.5$) can improve 0.4% for 256-D unsupervised image retrieval, while Dropout can not improve dimension-constrained unsupervised image retrieval. Even though Dropout enforces partial features for classification, the global randomization within the mini-batch makes the optimization involve all feature dimensions. By contrast, the proposed random feature selection is fixed within the mini-batch thus it can benefit from optimization in a sub-feature space.

## 5 CONCLUSIONS

This paper introduces Unicom, a simple yet effective framework for universal and compact feature embedding. Given the automatically clustered large-scale data, we employ one random negative class selection to improve the robustness under the heavy inter-class conflict and another random feature selection to improve the compactness of the feature representation. For both unsupervised and supervised image retrieval on different datasets, the proposed Unicom achieves state-of-the-art performance, confirming that cluster discrimination is beneficial to explore the semantic structure within the large-scale training data.

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

# A  APPENDIX

## A.1  MODEL ARCHITECTURES

We follow the same architecture design as CLIP. Tab. 7 describes the details of architectures.

## A.2  VISUALIZATION OF PSEUDO CLUSTERS AND DATA DISTRIBUTION

In Fig. 4, we show the data distribution under different settings of class number $k$. In Fig. 5, we show some exemplar classes from the proposed automatic clustering. As we can see, there are some fine-grained classes, such as the top with love icons and the top with cartoons. Even though such clustering is reasonable and explainable, there is class confusion if we classify these samples from other views, such as color and targeting customer age.

Table 7: The architecture parameters for ViT models.

| Model | Batch Size (128 V100) | FLOPs G | Embedding dimension | Input resolution | Vision Transformer layers | width | heads |
|---|---|---|---|---|---|---|---|
| ViT-B/32 | 256*128 | 4.3 | 512 | 224 | 12 | 768 | 12 |
| ViT-B/16 | 256*128 | 17.6 | 768 | 224 | 12 | 768 | 12 |
| ViT-L/14 | 64*128 | 80.9 | 768 | 224 | 24 | 1024 | 16 |
| ViT-L/14-336 | 48*128 | 191.3 | 768 | 336 | 24 | 1024 | 16 |

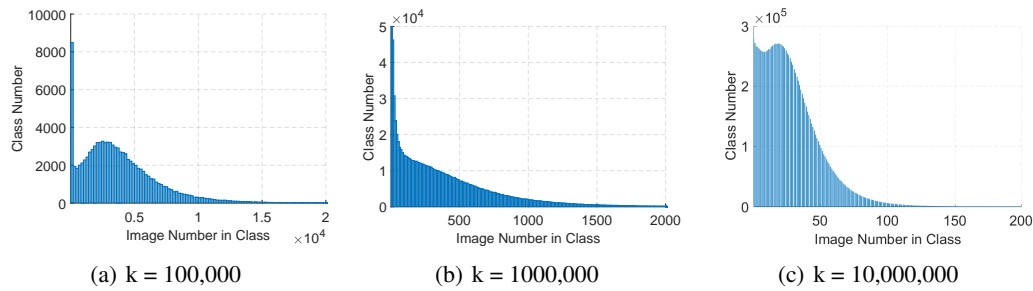

(a) k = 100,000  (b) k = 1000,000  (c) k = 10,000,000

Figure 4: Data distribution under different settings of cluster number $k$.

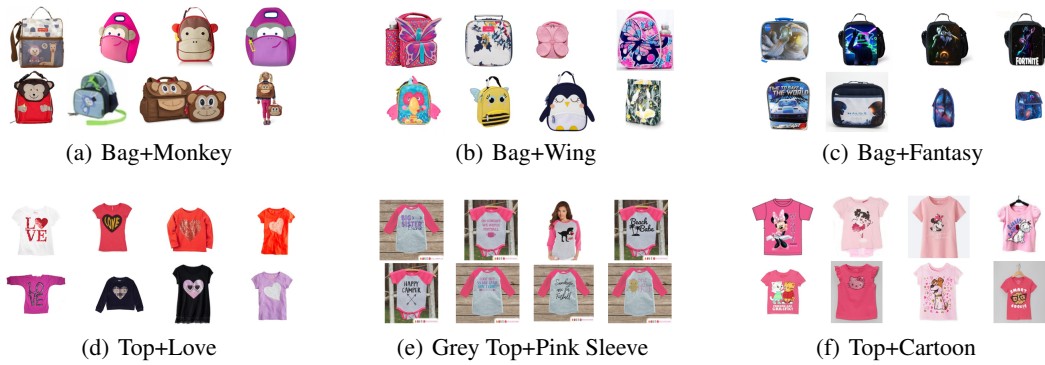

(a) Bag+Monkey  (b) Bag+Wing  (c) Bag+Fantasy

(d) Top+Love  (e) Grey Top+Pink Sleeve  (f) Top+Cartoon

Figure 5: Pseudo classes clustered by the image encoder and text encoder of the pre-trained CLIP model. The class name is given based on the manual observation of images and texts.

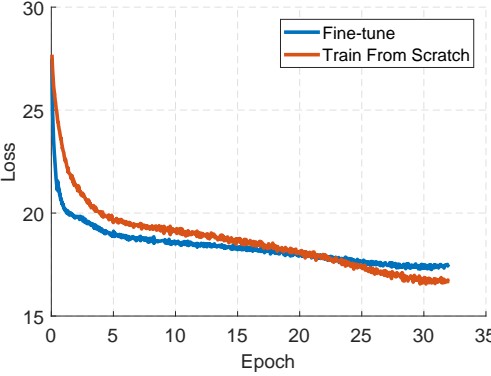

Figure 6: Training loss curves of fine-tuning from the CLIP model and training from scratch.

Table 8: Comparison between fine-tuning and training from scratch. Top-1 accuracy(%) of linear probe is reported on 13 image classification datasets. ViT B/32 is used here.

| | CIFAR10 | CIFAR100 | Caltech101 | Cars | Flowers | Food101 | Birdsnap | SUN397 | DTD | Aircraft | Pets | EuroSAT | ImageNet | Average |
|---|---|---|---|---|---|---|---|---|---|---|---|---|---|---|
| ViT B/32 (Scratch) | 96.8 | 86.6 | 94.6 | 93.3 | 98.5 | 85.8 | 70.2 | 74.6 | 78.0 | 70.7 | 93.1 | 96.8 | 75.0 | 85.7 |
| ViT B/32 (Fine-tune) | 95.8 | 83.3 | 94.1 | 92.4 | 98.5 | 87.3 | 67.3 | 75.3 | 79.8 | 66.5 | 92.8 | 96.1 | 75.1 | 85.0 |

Table 9: Object detection and instance segmentation on COCO. We evaluate bounding-box AP ($AP^{bb}$) and mask AP ($AP^{mk}$) on val2017.

| Method | Pre-training Data | $AP^{bb}$ | $AP^{bb}_{50}$ | $AP^{bb}_{75}$ | $AP^{mk}$ | $AP^{mk}_{50}$ | $AP^{mk}_{75}$ |
|---|---|---|---|---|---|---|---|
| ViT B/16 (Li et al., 2022) | IN-1K | 47.6 | - | - | 42.4 | - | - |
| ViT B/16 (Li et al., 2022) | IN-21K | 47.8 | - | - | 42.6 | - | - |
| ViT B/16 OPEN-CLIP | LAION 400M | 48.1 | 69.1 | 51.7 | 42.6 | 66.4 | 44.3 |
| ViT B/16 Ours | LAION 400M | 48.5 | 69.8 | 52.4 | 42.9 | 66.9 | 45.9 |

## A.3 TRAINING FROM SCRATCH VS. FINE-TUNING FROM THE CLIP MODEL

In this paper, the CLIP model is only used for the clustering step and our models are trained from scratch. In Fig. 6, we compare the training loss curves between fine-tuning and training from scratch. For fine-tuning, the backbone is initialized from the CLIP model (ViT-B/12), and the classifier (FC layer) is randomly initialized. The fine-tuning strategy can converge faster than training from scratch, but the final loss value is higher. In Tab. 8, we also find that training from scratch outperforms fine-tuning from the CLIP model by $0.7\%$ on the task of linear probe.

## A.4 COMPARISONS ON COCO DETECTION AND SEGMENTATION

Following the experiment setting in (Li et al., 2022), we use Mask R-CNN (He et al., 2017) for bounding-box object detection and instance segmentation. We fine-tune models on the COCO (Lin et al., 2014) train2017 split and evaluate on the val2017 split. In Tab. 9, our method outperforms both OPEN-CLIP and supervised pre-training in all metrics, demonstrating the effectiveness of the proposed cluster discrimination.

## A.5 LINEAR PROBE DATASETS

We use 13 image classification datasets to prove the effectiveness of our method. These datasets include CIFAR10(Krizhevsky & Hinton, 2009), CIFAR100(Krizhevsky & Hinton, 2009), Caltech101(Fei-Fei et al., 2004), Stanford Cars(Krause et al., 2013a), Oxford Flowers(Nilsback & Zisserman, 2008), Food-101(Bossard et al., 2014), Birdsnap(Berg et al., 2014), SUN397(Xiao et al., 2010), Describable Textures(Cimpoi et al., 2014), FGVC Aircraft(Maji et al., 2013), Oxford-IIIT Pets(Parkhi et al., 2012), EuroSAT(Helber et al., 2019), ImageNet-1k(Russakovsky et al., 2015). Details on each dataset and the corresponding evaluation metrics are provided in Tab. 10.

## A.6 IMAGE RETRIEVAL DATASETS

The training and evaluation of image retrieval experiments on seven widely used datasets, namely CUB-200-2011(CUB) (Welinder et al., 2010), Stanford Cars(Cars196) (Krause et al., 2013b), Stanford Online Products(SOP) (Oh Song et al., 2016), In-shop Clothes Retrieval(In-Shop) (Liu et al., 2016b), iNaturalist (Van Horn et al., 2018), VehicleID (Liu et al., 2016a), and Google Landmarks dataset (GLDv2) (Weyand et al., 2020). The number of examples and classes can be found in Tab. 11.

Table 10: List of linear probe datasets with the data distribution and evaluation metrics.

| Dataset | Classes | Train size | Test size | Evaluation metric |
|---|---|---|---|---|
| CIFAR-10 | 10 | 50,000 | 10,000 | accuracy |
| CIFAR-100 | 100 | 50,000 | 10,000 | accuracy |
| Caltech-101 | 102 | 3,060 | 6,085 | mean-per-class |
| Stanford Cars | 196 | 8,144 | 8,041 | accuracy |
| Oxford Flowers | 102 | 2,040 | 6,149 | mean per class |
| Food-101 | 102 | 75,750 | 25,250 | accuracy |
| Birdsnap | 500 | 42,283 | 2,149 | accuracy |
| SUN397 | 397 | 19,850 | 19,850 | accuracy |
| Describable Textures | 47 | 3,760 | 1,880 | accuracy |
| FGVC Aircraft | 100 | 6,667 | 3,333 | mean per class |
| Oxford-IIIT Pets | 37 | 3,680 | 3,669 | mean per class |
| EuroSAT | 10 | 10,000 | 5,000 | accuracy |
| ImageNet | 1000 | 1,281,167 | 50,000 | accuracy |

Table 11: Dataset composition for training and evaluation in the image retrieval task.

| Dataset | Images | Classes |
|---|---|---|
| CUB Train (Welinder et al., 2010) | 5,864 | 100 |
| CUB Test  (Welinder et al., 2010) | 5,924 | 100 |
| Cars196 Train (Krause et al., 2013b) | 8,054 | 98 |
| Cars196 Test (Krause et al., 2013b) | 8,131 | 98 |
| SOP Train (Oh Song et al., 2016) | 59,551 | 11,318 |
| SOP Test (Oh Song et al., 2016) | 60,502 | 11,316 |
| In-Shop (Liu et al., 2016b) | 25,882 | 3,997 |
| In-Shop (Liu et al., 2016b) | 26,830 | 3,985 |
| iNaturalist Train (Van Horn et al., 2018) | 325,846 | 5,690 |
| iNaturalist Test (Van Horn et al., 2018) | 136,093 | 2,452 |
| VehicleID Train (Liu et al., 2016a) | 110,178 | 13,134 |
| VehicleID Test (Liu et al., 2016a) | 40,365 | 4,800 |
| GLDv2 Train (Weyand et al., 2020) | 1,580,470 | 81,314 |
| GLDv2 Test  (Weyand et al., 2020) | 762,884 | 1,129 |

