# OpenReview forum: "Unicom: Universal and Compact Representation Learning for Image Retrieval"
_ICLR.cc/2023/Conference — ICLR 2023 poster_

### Official Review · Reviewer_HyPg · 2022-10-19

**Confidence:** 4
**Correctness:** 3
**Technical Novelty And Significance:** 2
**Empirical Novelty And Significance:** 3
**Recommendation:** 8

**Clarity, Quality, Novelty And Reproducibility:**

Clarity: The paper is generally clear and the reader can understand what the authors are proposing. However, the story is not very clear/consistent, as discussed above.

Quality: The idea of improving cluster discrimination via random negative prototype selection is shown to work clearly, and is simple. Experiments are well done. However, I see several issues as listed in the weaknesses above.

Novelty: Novelty on improving cluster discrimination as mentioned above. The random feature subsampling is also novel, but doesn’t sound well motivated to me.

Reproducibility: The approach is generally simple, I don’t foresee reproducibility issues. The authors also promise to release code and models to ensure reproducibility.

**Strength And Weaknesses:**

Strengths:

S1) A well-motivated idea to improve unsupervised learning via cluster discrimination. Cluster discrimination may suffer from inter-class conflicts of the prototypes, and the authors propose a very simple and efficient method to deal with it: randomly sample the negative prototypes. Ablation experiments show large improvements when using this idea.

S2) The proposed pre-training method outperforms other CLIP training methods on image classification linear probe experiments and on unsupervised image retrieval experiments. The comparison is strict against the OPEN-CLIP work, where the exact same dataset is used.

S3) The results on the supervised image retrieval tasks outperform other recent competitor methods.

S4) Overall, the experiments are comprehensive, including ablations and even extra comparisons on GLDv2 and the recent Google Universal Embedding Challenge.

Weaknesses:

W1) Random selection of class prototypes seems very simple and effective, but I am wondering whether something smarter can be even better. For example, discarding “hard negative prototypes” directly instead of only sampling randomly. The paper does not provide such comparisons, which would make the proposal stronger.

W2) I wonder if it is fair to call the proposed method “unsupervised”. The technique works on a large dataset of images and associated text, which could be considered already some form of weak supervision (it has more details than the Instagram hashtag dataset, for example). The CLIP text embeddings are even directly used to perform clustering. Generally, “unsupervised” methods work directly on images only and nothing else.

W3) I don’t see a direct connection between equations (1) and (2) in section 3.1. How is equation (2) actually maximizing the log-likelihood function? Do you mean that p(x_i;\theta) would be the probability of drawing the augmented image? Please explain. Similar comment for equations (3) and (4).

W4) Figure (3)-a is confusing. By “class number”, do the authors mean the ID of the class? By “photo number”, do you mean the number of photos in that class?

W5) There is a very straightforward connection between the random feature subsampling and the concept of Dropout. This connection is not made, but it should be.

W6) The random feature subsampling method works comparably to PCA or the addition of an FC layer to reduce dimensionality in the network. I don’t see a concrete benefit of using it, compared to a simple FC layer to reduce dimensionality, which is more principled.

W7) The story of the paper is inconsistent. There are many experiments on image classification datasets (section 4.2), but the paper is directed at the image retrieval problem.

Minor details:

M1) In the caption of Fig 1: the word “dataset” is repeated on the second line.

M2) Page 3: “cluttering” -> “clustering”

M3) Section 3.1: the set X  is repeated twice in the same first sentence, please remove one of them to make it more concise.


**Summary Of The Paper:**

The paper proposes a method to learn representations for image retrieval with two main ideas: 1) during pre-training, automatically clustering a large image-text dataset and randomly selecting inter-class prototypes to avoid noisy assignments, and 2) to reduce embedding dimensionality, randomly select feature dimensions to be deactivated. Experiments are performed on standard datasets in the area of image retrieval, besides on standard image classification datasets, comparing against unsupervised and supervised methods.

**Summary Of The Review:**

This is a paper with interesting ideas, some of which are simple and shown to work well, for an important problem in computer vision. Experiments are comprehensive. However, I also see several issues at this point, as described in the weaknesses above.

---

> ### Author Response · Authors · 2022-11-19
> **Author's Response to Reviewer HyPg (Part-1 out of 2)**
>
> We thank the reviewer for the detailed and constructive comments to improve our paper.
> We are glad that you think our ideas work well for an important problem in computer vision and our experiments are comprehensive.
> Below, we address your concerns on weaknesses point-by-point.
>
> **Q1**: discarding “hard negative prototypes” directly instead of only sampling randomly
>
> **A1**: We consider a threshold ($\tau$) to drop noisy negative prototypes during training.
>
> | Tasks | $r_1=0.1$ | $\tau=0.4 $ | $\tau=0.5 $ | $\tau=0.6 $ |$\tau=0.7 $ | $\tau=0.8 $|
> | :----: | :----:    |:----:       |:----:       |:----:       | :----:     | :----:     |
> | Linear Probe  | 85.7    | 79.9     | 83.1   | 84.4   | 84.2  | 83.7 |
> | Unsup. Retr.  | 62.2    | 56.4     | 59.4   | 61.0   | 61.4  | 60.8 |
>
> As the threshold $\tau$ increases from $0.4$ to $0.8$, the performance raises first and then drops.
> When the threshold is set to $0.6$, the online hard negative prototype discarding method obtains the best results
> of $84.4\%$ and $61.0\%$ on the tasks of the linear probe and unsupervised image retrieval.
> By contrast, the proposed random negative prototype selection achieves $85.7\%$ and $62.2\%$ when the random class selection ratio is set as $0.1$.
> We put this experiment in the revised Appendix A.7.
>
> **Q2**: I wonder if it is fair to call the proposed method “unsupervised”
>
> **A2**: Compared to supervised methods, our method does not use manual annotations. However, the proposed method relies on the pre-trained CLIP model, thus it is weakly supervised.
> We changed the descriptions accordingly in the revision.
> For the image retrieval tasks, if no label from the retrieval dataset is used, it is called unsupervised image retrieval even if the backbone is initialized from the ImageNet pre-trained network[1].
> If the labels from the retrieval datasets are used for fine-tuning, it is called supervised image retrieval.
>
> [1]Kim, Sungyeon, et al. Self-Taught Metric Learning without Labels. CVPR 2022.
>
> **Q3**: I don’t see a direct connection between equations (1) and (2) in section 3.1.
> How is equation (2) actually maximizing the log-likelihood function?
> Do you mean that $p(x_i;\theta)$ would be the probability of drawing the augmented image?
>
> **A3**: In the revision, we deleted the equation of maximizing the log-likelihood to make the descriptions more straightforward and save space in the main paper.
> $p(x_i;\theta)$ and $p(x_i,w_i;\theta)$ refer to the probability drawing the augmented image feature (positive sample)
> and the corresponding centroid (target/positive prototype), respectively.
>
> **Q4**: Figure (3)-a is confusing. By “class number”, do the authors mean the ID of the class?
> By “photo number”, do you mean the number of photos in that class?
>
> **A4**: Figure 3 gives the data distribution after our one-step offline clustering. The X label is the photo number, and the Y label is the class frequency.
> At the left of the figure, there are lots of classes with a small number of images. Af the right of the figure, there are a few classes with a large number of images.
> In the revision, we moved the data distribution to Appendix A.2.

---

> > ### Author Response · Authors · 2022-11-19
> > **Author's Response to Reviewer HyPg (Part-2 out of 2)**
> >
> > **Q5**: The connection to Dropout is not made, but it should be.
> >
> > **A5**: Our random feature selection $\Gamma_t$ is same for all training samples within the mini-batch.
> > $\Gamma_t$ is applied to both feature $e_i$ and prototypes $w_j$, thus the dimension of gradients decreases to $d'$.
> > By contrast, Dropout is independently applied to each individual feature $e_i$ within the mini-batch
> > by setting a specific ratio $r_3\in\left [0,1 \right ]$ of neurons to $0$ and enlarging the rest neurons by $1/(1-r_3)$.
> > The dimension of gradients is still $d$. Dropout is designed for regularization.
> > Our random feature selection is designed to enhance the representation power of each neuron from the 512-D feature vector.
> >
> > | Tasks  | $r_2=1.0 $ |$r_2=0.5$ | $r_3=0.25 $ | $\tau=0.5 $ |
> > | :----: | :----:    |:----:       |:----:       | :----:       |
> > | Linear Probe  | 85.7 | 85.5    | 85.4     | 85.1   |
> > | Unsup. Retr.  | 62.2 | 62.1    | 62.1     | 61.9   |
> > | Unsup. Retr. $^{128}$| 58.2 | 59.1    | 57.4     | 56.9   |
> > | Unsup. Retr. $^{64}$ | 53.9 | 55.1    | 53.4     | 53.1   |
> >
> > $r_2$ is the random feature selection ratio and $r_3$ is the Dropout ratio.
> > As we can see from this Table, both random feature selection and Dropout can not improve linear probe and unsupervised image retrieval
> > at a full dimension of $512$ as the LAION 400M dataset is large enough and regularization is not necessary for the final classification layer.
> > In addition, the proposed random feature selection ($r_2=0.5$) can improve 0.9\% and 1.2\% for 128-D and 64-D unsupervised image retrieval,
> > while Dropout can not improve dimension-constrained unsupervised image retrieval.
> > Even though Dropout enforces partial features for classification,
> > the global randomization within the mini-batch makes the optimization involve all feature dimensions.
> > By contrast, the proposed random feature selection is fixed within the mini-batch, thus it can benefit from optimization in a sub-feature space.
> >
> > In the revision, we compared with Dropout in the method and experiment.
> >
> > **Q6**: The random feature subsampling method works comparably to PCA or the addition of an FC layer to reduce dimensionality in the network.
> > I don’t see a concrete benefit of using it, compared to a simple FC layer to reduce dimensionality, which is more principled.
> >
> > **A6**: In Table 7, the representation produced by PCA is sub-optimal because it is a post-processing step,
> > while the FC layer and the proposed random feature sampling can be trained in an end-to-end way.
> >
> > When the random feature selection ratio is set to $0.5$, the proposed Unicom consistently surpasses the FC baseline across different dimensions.
> > For example, under the test dimension of 32, Unicom ($r_2 = 0.5$) achieves 82.1\%, while the FC layer only achieves 81.7\% on the CUB dataset.
> > For the low-dimension test scenario(e.g., 32-D), the solution based on the FC projector is hard to train.
> > By contrast, the proposed method still keeps all 512-D features in the memory.
> > Besides, the best random feature dimension is also higher than the dimension required by the test scenario.
> > For example, our method achieves the best performance for the 32-D test scenario
> > when the random feature selection ratio is set to $0.5$ and 256-D features are selected in each iteration for training.
> > In each iteration, our method randomly selects a fixed ratio of features to construct the softmax loss to enhance the representation power of subset neurons from the 512-D feature vector.
> >
> > Please note that the FC projectors need to be trained under each dimension constraint while our Unicom is only trained once and tested across different dimensions. In Appendix A.9, we also verified the effectiveness of the proposed Unicom on the Google Universal Image Embedding competition.
> >
> > **Q7**: The story of the paper is inconsistent.
> > There are many experiments on image classification datasets (section 4.2), but the paper is directed at the image retrieval problem.
> >
> > **A7**: In the revision, we moved the fine-tuning experiments on ImageNet and COCO to Appendix.
> > We unified the whole story as universal feature embedding.
> > For zero-shot learning tasks, we have the experiments on the linear probe and unsupervised image retrieval.
> > For transfer learning tasks, we have experiments on supervised image retrieval.
> > The structure of this paper is then more clear. We also fixed all typos you mentioned.
> >
> > **We are pleased for your appreciation on our paper. we hope you feel the flaws are addressed in our response.**

---

> > > ### Comment · Reviewer_HyPg · 2022-12-05
> > > **Thank you for the rebuttal**
> > >
> > > I'd like to thank the authors for responding to all my comments/questions. I really appreciate the new results when answering to Q1 and Q5, and their inclusion in the paper. This makes the paper much more complete overall. Also the revision of the story (Q7) makes it more consistent.
> > >
> > > Given all the changes, I have updated my rating to "accept".

---

### Official Review · Reviewer_FUwi · 2022-10-23

**Confidence:** 4
**Correctness:** 4
**Technical Novelty And Significance:** 2
**Empirical Novelty And Significance:** 3
**Recommendation:** 5

**Clarity, Quality, Novelty And Reproducibility:**

Clarity: Overall paper is well written. Only non-clear part is that, during softmax learning, whether the entire image encoder is learned from scratch, or fine-tuned from the pre-existing CLIP image embedding. Would be good to compare the two in ablation studies.

Novelty: low (see above)

Reproducibility: approach is very intuitive and easy to reproduce

**Strength And Weaknesses:**

Strength
+ Intuitive idea and easy to reproduce
+ Extensive experimental results + ablation studies

Weakness
- Novelty is a bit low. Sampled softmax is a commonly used technique.
- Dimension sampling is a bit strange. A more natural approach to me would be to directly learn a projection to low dimensional feature space.

**Summary Of The Paper:**

This paper presents a self-supervised image representation approach that leverages using pre-existing CLIP image-text embedding. The approach first clusters images using CLIP embedding, and then learns image feature with softmax loss using cluster ids as class ids. In addition, random dimension sampling is employed during learning to achieve low dimensionality for retrieval tasks. The authors conducted extensive results on benchmark datasets to demonstrate the effectiveness of proposed approach in various settings.

**Summary Of The Review:**

While approach is appealing and practically useful, the novelty seems slightly below the bar of ICLR.  In additional, the dimension sampling part seems unnecessary to me. I feel directly learning a feature projection would be more intuitive and work better.

---

> ### Author Response · Authors · 2022-11-19
> **Author's Response to Reviewer FUwi**
>
> We thank the reviewer for the constructive comments to improve our paper.
> We are glad that you think our idea is intuitive to reproduce and our experiments are extensive.
> Below, we address your concerns on weaknesses point-by-point.
>
> **Q1**: Novelty is a bit low. Sampled softmax is a commonly used technique.
>
> **A1**: As reviewer htKd said, this idea, although simple (this also makes it widely applicable) is novel in this context.
> The sampled softmax applied in our context is designed to
> (1) solve heavy inter-class conflict from the single pre-step of off-line clustering, and
> (2) enhance the representation power of each neuron thus facilitating dimension-constrained image retrieval.
> Our conflict-robust cluster discrimination method can significantly boost the semantic embedding power of the CLIP model.
> The proposed Unicom outperforms state-of-the-art unsupervised and supervised image retrieval approaches on multiple benchmarks
> with substantial improvement under different dimension constraints.
>
> **Q2**: Dimension sampling is a bit strange.
> A more natural approach to me would be to directly learn a projection to low dimensional feature space.
>
> **A2**: In Table 7, the proposed Unicom consistently surpasses the FC baseline across different test dimensions
> when the random feature selection ratio is set to $0.5$.
> For example, under the test dimension of 32, Unicom ($r_2 = 0.5$) achieves 82.1\%,
> while the FC layer only achieves 81.7\% on the CUB dataset.
> For the low-dimension test scenario (e.g., 32-D), the solution based on the FC projector is hard to train.
> By contrast, the proposed method still keeps all 512-D features in the memory.
> Besides, the best random feature dimension is also higher than the dimension required by the test scenario.
> For example, our method achieves the best performance for the 32-D test scenario
> when the random feature selection ratio is set to $0.5$ and 256-D features are selected in each iteration for training.
> In each iteration, our method randomly selects a fixed ratio of features to construct the softmax loss to enhance the representation power of subset neurons from the 512-D feature vector.
>
> Please note that the FC projectors need to be trained under each dimension constraint while our Unicom is only trained once and tested across different dimensions. In Appendix A.9, we also verified the effectiveness of the proposed Unicom on the Google Universal Image Embedding competition.
>
> **Q3**: During softmax learning, whether the entire image encoder is learned from scratch,
> or fine-tuned from the pre-existing CLIP image embedding. Would be good to compare the two in ablation studies.
>
> **A3**: In this paper, all our models are trained from scratch for 32 epochs.
> In the revised Appendix A.5, we add a comparison between fine-tuning from the CLIP weights and training from scratch.
> For fine-tuning, the backbone is initialized from the CLIP model (ViT-B/12), and the classifier (FC layer) is randomly initialized.  As we can see from the training loss, the fine-tuning strategy can converge faster than training from scratch, but the final loss value is higher.
> In addition, training from scratch outperforms fine-tuning from the CLIP weights by 0.7\% on the task of the linear probe.
> Therefore, we choose to train from scratch in this paper.
>
> **We hope your concerns regarding this paper are addressed in our response.**

---

### Official Review · Reviewer_1Pq2 · 2022-10-24

**Confidence:** 4
**Correctness:** 3
**Technical Novelty And Significance:** 2
**Empirical Novelty And Significance:** 2
**Recommendation:** 5

**Clarity, Quality, Novelty And Reproducibility:**

The novelty of the paper is not significant. The sampling dimension of the feature is not too much different from Dropout method[1], which has been widely used in CV/ML pipelines.



[1] Srivastava, Nitish, et al. "Dropout: a simple way to prevent neural networks from overfitting." The journal of machine learning research 15.1 (2014): 1929-1958.


**Strength And Weaknesses:**

The method is easy to understand and implement. The experiments about using VL models for zero-shot/transfer learning tasks are exhaustive.


**Summary Of The Paper:**

This paper proposes to use random sampling for samples in clustering classes to avoid inter-class conflict and a random dropout mechanism for features to generate compact features.

**Summary Of The Review:**

As I mentioned above, although the method is easy to implement, the novelty of this paper is not significant. Especially for feature random selection part. For class selection, although it is easy to implement, it lacks further detailed analysis and improvement. For instance, we can use another clustering method for each cluster to exclude the potential inter-class conflict.

---

> ### Author Response · Authors · 2022-11-19
> **Author's Response to Reviewer 1Pq2**
>
> We thank the reviewer for the constructive comments to improve our paper.
> We are glad that you think our method is easy to understand and implement and our experiments are exhaustive.
> Below, we address your concerns on weaknesses point-by-point.
>
> **Q1**: The novelty of the paper is not significant.
> The sampling dimension of the feature is not too much different from Dropout method, which has been widely used in CV/ML pipelines.
>
> **A1**: Dropout is a regularization technique designed for over-fitting alleviation on small training data.
> The proposed random feature selection is designed to enhance the representation power of each neuron from the 512-D feature vector.
> In the revision, we compared with Dropout in the method and experiment.
>
> Our random feature selection $\Gamma_t$ is same for all training samples within the mini-batch.
> $\Gamma_t$ is applied to both feature $e_i$ and prototypes $w_j$, thus the dimension of gradients decreases to $d'$.
> By contrast, Dropout is independently applied to each individual feature $e_i$ within the mini-batch
> by setting a specific ratio $r_3\in\left [0,1 \right ]$ of neurons to $0$ and enlarging the rest neurons by $1/(1-r_3)$.
> The dimension of gradients is still $d$.
>
> | Tasks  | $r_2=1.0 $ |$r_2=0.5$ | $r_3=0.25 $ | $\tau=0.5 $ |
> | :----: | :----:    |:----:       |:----:       |:----:       |
> | Linear Probe  | 85.7 | 85.5    | 85.4     | 85.1   |
> | Unsup. Retr.  | 62.2 | 62.1    | 62.1     | 61.9   |
> | Unsup. Retr. $^{128}$| 58.2 | 59.1    | 57.4     | 56.9   |
> | Unsup. Retr. $^{64}$ | 53.9 | 55.1    | 53.4     | 53.1   |
>
> $r_2$ is the random feature selection ratio and $r_3$ is the Dropout ratio.
> As we can see from this Table, both random feature selection and Dropout can not improve linear probe and unsupervised image retrieval
> at a full dimension of $512$ as the LAION 400M dataset is large enough and regularization is not necessary for the final classification layer.
> In addition, the proposed random feature selection ($r_2=0.5$) can improve 0.9\% and 1.2\% for 128-D and 64-D unsupervised image retrieval,
> while Dropout can not improve dimension-constrained unsupervised image retrieval.
> Even though Dropout enforces partial features for classification,
> the global randomization within the mini-batch makes the optimization involve all feature dimensions.
> By contrast, the proposed random feature selection is fixed within the mini-batch, thus it can benefit from optimization in a sub-feature space.
>
> **Q2**: For class selection, although it is easy to implement, it lacks further detailed analysis and improvement.
> For instance, we can use another clustering method for each cluster to exclude the potential inter-class conflict.
>
> **A2**: In the revision, we add detailed analysis regarding the proposed random inter-class selection
> by gradient analysis on the embedding feature (Eq. 5).
> In the revised Appendix A.7, we also compared the proposed random negative prototype selection
> with a thresholding method to discard noisy negative prototypes.
> The experimental results indicate that the proposed random negative prototype selection can achieve better performance than the thresholding method.
>
> Achieving good clustering results on 400M images by using one single pre-step of off-line clustering is an extremely challenging task.
> No matter how advanced a clustering method is, there are intra-class and inter-class noises. It is inevitable.
> We discussed the cluster number in the section of the ablation study.
> The cluster number can balance the intra-class noises and inter-class noises.
> Too small cluster numbers will incur heavy intra-class noise, which dramatically decreases the performance of the pre-trained classification model.
> Besides, too many clusters will increase the inter-class noise ratio and undermine the discriminative power of the pre-trained model.
>
> In this paper, we are not focusing on improving clustering.
> We only employ one step of off-line clustering but design a robust classifier to achieve good feature representation
> when training on the automatically clustered large-scale data.
> Inspired by your suggestion, we will explore how to ensemble features or clustering methods to alleviate inter-class conflicts in the future.
>
> **We hope your concerns regarding this paper are addressed in our response.**

---

### Official Review · Reviewer_htKd · 2022-10-29

**Confidence:** 4
**Correctness:** 4
**Technical Novelty And Significance:** 3
**Empirical Novelty And Significance:** 3
**Recommendation:** 6

**Clarity, Quality, Novelty And Reproducibility:**

- The paper is well written and easy to understand.
- This code is easily reproducible because the idea is simple, but the experiments only if you have access to 128 V100 GPUs
- The idea of subsampling classes and features might not be novel in general, but it is when applied to this context.

**Strength And Weaknesses:**

Pros
- The idea and motivation behind the method is presented very well by pointing out the limitations of currents methods regarding their clustering approach and proposing a possible solution.
- This idea is also very simple, which makes it widely applicable, not only to this problem/setting.
- Experiments are also extensive and thorough.

Cons:
- The fact that you need access to a large pre-trained model it makes its applicability more limiting and not directly comparable to OPEN-CLIP. I see the approach as a way of further improving the results of a given base model, CLIP in this case.
- Related to my previous point, and given that we have assume that we have access to CLIP and that we are going to train the same backbone architecture (is this true?), would it make sense to initialize the backbone using CLIP's weights? Would this speed up training without impacting performance? This might address the problem of needing to have access to a large cluster of GPUs.
- It would have been interesting to see this applied to other large pre-trained models used as base, eg DINO or MAE.
- In table 8, I did not understand what are the different columns below Unicom. Could the authors elaborate please?

**Summary Of The Paper:**

This paper addresses the problem of representation learning for image retrieval and follow-ups on recent self-supervised methods that use cluster discrimination to handle the limitations of instance discrimination. These methods however either require to perform iterative clustering, or online clustering to avoid multiple iterations over the entire dataset but that still suffer from the collapsing problem. The authors propose a method that performs random selections along class and feature to alleviate the inter-class conflict and to improve feature compactness that only requires a single pre-step of off-line clustering.

**Summary Of The Review:**

The paper is well written and the idea and motivation behind the method is presented very well. This idea, although simple (this also makes it widely applicable) it is novel in this context. I also think that the methods has some limitations, like the fact that one assumes access to a large pre-trained model.

---

> ### Author Response · Authors · 2022-11-19
> **Author's Response to Reviewer htKd**
>
> We thank the reviewer for the detailed and constructive comments to improve our paper.
> We are glad that you think the idea and motivation are presented well, our idea is novel in this context and our experiments are extensive and thorough.
> Below, we address your concerns on weaknesses point-by-point.
>
> **Q1**: The method needs access to a large pre-trained model makes its applicability more limiting and not directly comparable to OPEN-CLIP.
>
> **A1**: In this paper, we aim at boosting the semantic embedding power of the CLIP model,
> which is trained by image-text contrastive learning.
> CLIP employs instance discrimination, which can not effectively capture the semantic information from the training data.
> To this end, we introduce a novel cluster discrimination approach, which can explore potential semantic structures in the training data.
> Since iterative clustering and discrimination can not be easily applied to 400M images,
> a large pre-trained model (e.g., the CLIP model) is needed for one step of offline clustering.
>
> Open-CLIP is a reimplementation of image-text contrastive learning proposed by the CLIP paper on the publicly available LAION 400M dataset.
> By using the CLIP model for clustering, the average linear probe accuracy of our ViT B/32 is 85.7%.
> By using the OPEN-CLIP model for clustering, the average linear probe accuracy of our ViT B/32 is 84.1%.
> Both results are obviously better than the original result (82.1%) of OPEN-CLIP ViT B/32, indicating the effectiveness of the proposed cluster discrimination.
>
>
> | Tasks | CLIP Clustering + Our Method| OPEN-CLIP Clustering + Our Method | OPEN-CLIP Baseline|
> | :----: | :----:    |:----:       |:----:       |
> | Linear Probe  | 85.7    | 84.1    |  82.1|
> | Unsup. Retr.  | 62.2    | 60.9    |  55.0|
>
> **Q2**: the same backbone architecture is trained? would it make sense to initialize the backbone using CLIP's weights?
> Would this speed up training without impacting performance? This might address the problem of needing to have access to a large cluster of GPUs.
>
> **A2**: For a fair comparison, all ViT models (i.e., ViT B/32, ViT B/16, and ViT L/14) in our experiments follow the same architecture designs as in CLIP,
> and all our models are trained from scratch for 32 epochs.
> In the revised Appendix A.5, we add a comparison between fine-tuning from the CLIP weights and training from scratch.
> For fine-tuning, the backbone is initialized from the CLIP model (ViT-B/12), and the classifier (FC layer) is randomly initialized.  As we can see from the training loss, the fine-tuning strategy can converge faster than training from scratch, but the final loss value is higher.
> In addition, training from scratch outperforms fine-tuning from the CLIP weights by 0.7\% on the task of the linear probe.
> Therefore, we choose to train from scratch in this paper.
>
> **Q3**: apply to other large pre-trained models used as the base, eg DINO or MAE.
>
> **A3**: The proposed cluster discrimination can also improve MAE. However, the baseline of MAE is too low on unsupervised image retrieval.
> Therefore, we focus on improving the semantic embedding power of the CLIP model to achieve state-of-the-art performance on the image retrieval task.
> In the revised Appendix A.6, we added experiments using MAE.
>
> | Tasks | MAE | MAE Clustering + ViT B/32 Discrimination | CLIP Clustering + ViT B/32 Discrimination |
> | :----: | :----:    |:----:  | :----:  |
> | Linear Probe  |   76.1| 78.3| 85.7|
> | Unsup. Retr.  |   34.9| 47.8| 62.2|
>
> **Q4**: In table 8, I did not understand what are the different columns below Unicom. Could the authors elaborate please?
>
> **A4**: For the proposed random feature selection, the feature dimension is always d (d=512) in memory, but in each iteration,
> only d’ (d’<512) feature values are randomly selected to construct the softmax loss.  d’ is the hyper-parameter for our compact feature representation learning. In the revision, we use the random feature selection ratio $r_2$ ($d’ = d*r_2$) to facilitate understanding.
>
> For example, $r_2$ = 0.5 (d’ = 256) means that we randomly select 256 feature values from the 512-D feature vector to construct the softmax loss in each iteration.
> There is a best range for $r_2$. When $r_2$ is too small, such as $r_2$ =0.0625 (d’=32), there is an obvious performance drop.
> After we select the best feature selection ratio for our Unicom, we train the Unicom model only once and test it across different dimensions.
> For example, if the test dimension is 64, we use the first 64-D features from 512-D features for testing.  If the test dimension is 128, we use the first 128-D features from 512-D features for testing.
>
> Please note that the test dimension does not need to be same as the training dimension d’.
> As we can see from the Table 7, when d’=256, we achieve the best result (84.2\%) for testing under the dimension of 64.
>
> **We are pleased for your appreciation on our paper. we hope you feel the flaws are addressed in our response.**

---

### Public Comment · ~Yash_Patel2 · 2023-05-14
**Are the empirical comparisons fair?**

Hello,

Many thanks for sharing your interesting work. I noticed that the projection head of your models is substantially bigger than SWAG (Singh et al., CVPR 2022), OpenCLIP models and Timm's implementation of ViT that is used in recall@k surrogate (Patel et al., CVPR 2022). I ran a quick parameter counter for these models following the RS@k implementation, that is, with a layer norm and linear projection. Here are the number of parameters for each model:

```
ViT-B/32 Timm: 87850496
ViT-B/32 CLIP: 87849728
ViT-B/32 UNICOM: 117118464
ViT-B/16 Timm: 86193920
ViT-B/16 CLIP: 86193152
ViT-B/16 UNICOM: 202363136
ViT-B/16 SWAG: 86193920
```

It is clear that the UNICOM model has substantially higher number of parameters than the baselines used for the comparison. The additional parameters are from the projection head (model.feature in the implementation). With this in mind, are the comparisons with RS@k (Patel et al., CVPR 2022), SWAG (Singh et al., CVPR 2022) and CLIP fair?

---

> ### Author Response · Authors · 2023-05-14
> **The projection head structure is following ArcFace.**
>
> In relation to the projection head structure utilized in our Vision Transformer (ViT) model, it's worth noting that its origin can be traced back to the arcface paper.
>
> We're in the process of updating our experiment results on Github, which will feature a projection head structure akin to that of CLIP. Based on our preliminary tests, we have been able to match performance metrics on a certain subset. This promising development leads us to further validation, particularly on the laion400m dataset. Stay tuned for more updates, and once again, thank you for your interest.

---

> > ### Public Comment · ~Yash_Patel2 · 2023-05-17
> > **Incorrect number of FLOPs in Table 3.**
> >
> > Note that these number of parameters also affect the FLOP counts that are in the paper (table 3).

---

### Decision · Program_Chairs · 2023-01-20

**Decision:**

Accept: poster

**Justification For Why Not Higher Score:**

relatively simple random sampling schemes that work well but not profound to be highlighted yet

**Justification For Why Not Lower Score:**

good to share with the ICLR community as its implementation is easy and the approach is widely applicable.

**Metareview: Summary, Strengths And Weaknesses:**

The paper aims to improve the representations for image retrieval with automatic clustering of a large image-text dataset and randomly selecting inter-class prototypes to avoid noisy assignments, and reduction of embedding dimensionality by randomly selecting feature dimensions to be deactivated. Extensive experiments on standard image retrieval datasets, compared with unsupervised and supervised methods, have shown promising results. Overall the intuitive ideas and their rationale are presented well, and shown to work well in benchmarks.

**Note From Pc:**

if the above contains the word "oral" or "spotlight" please see: "oral" presentation means -> notable-top-5% and "spotlight" means -> notable-top-25%. As stated in our emails, we are disassociating presentation type from AC recommendations

**Summary Of Ac-Reviewer Meeting:**

Two reviewers score 8 and 6, and the other two gave 5, who are mainly concerned about novelty, commented that the ideas proposed are relatively simple. The authors have shown extensive benchmarks with improvement over SOTA methods and provided further analysis on both random prototype selection schemes. That is, the proposed method, though relatively simple, works well with good rationale.